# No evidence of SARS-CoV-2 in hospitalized patients with severe acute respiratory syndrome in five Italian hospitals from 1st November 2019 to 29th February 2020

Donatella Panatto[1,2]*, Andrea Orsi[1,2,3], Beatrice Marina Pennati[1], Piero Luigi Lai[1,2], Stefano Mosca[1], Bianca Bruzzone[3], Patrizia Caligiuri[3], Christian Napoli[4], Enrico Bertamino[5], Giovanni Battista Orsi[6], Ilaria Manini[1,7], Daniela Loconsole[8], Francesca Centrone[8], Elisabetta Pandolfi[9], Marta Luisa Ciofi Degli Atti[9], Carlo Concato[9], Giulia Linardos[9], Andrea Onetti Muda[9], Massimiliano Raponi[9], Livia Piccioni[9], Caterina Rizzo[9], Maria Chironna[8], Giancarlo Icardi[1,2,3]

1 Interuniversity Research Center on Influenza and Other Transmissible Infections (CIRI-IT), Genoa, Italy, 2 Department of Health Sciences, University of Genoa, Genoa, Italy, 3 Policlinico San Martino Hospital, Genoa, Italy, 4 Department of Medical Surgical Sciences and Translational Medicine, University La Sapienza, Rome, Italy, 5 Sant'Andrea Hospital, University La Sapienza, Rome, Italy, 6 Department of Public Health and Infectious Diseases, University La Sapienza, Rome, Italy, 7 Department of Molecular and Developmental Medicine, University of Siena, Siena, Italy, 8 Hygiene Section, Department of Biomedical Sciences and Human Oncology, University of Bari, Bari, Italy, 9 Bambino Gesù Children's Hospital, Rome, Italy

* panatto@unige.it

**Data Availability Statement:** All relevant data are within the paper and its Supporting Information files.

## Abstract

### Background

On 9th January 2020, China CDC reported a novel coronavirus (later named SARS-CoV-2) as the causative agent of the coronavirus disease 2019 (COVID-19).

Identifying the first appearance of virus is of epidemiological importance to tracking and mapping the spread of SARS-CoV-2 in a country. We therefore conducted a retrospective observational study to detect SARS-CoV-2 in oropharyngeal samples collected from hospitalized patients with a Severe Acute Respiratory Infection (SARI) enrolled in the DRIVE (Development of Robust and Innovative Vaccine Effectiveness) study in five Italian hospitals (CIRI-IT BIVE hospitals network) (1st November 2019 – 29th February 2020).

### Objectives

To acquire new information on the real trend in SARS-CoV-2 infection during pandemic phase I and to determine the possible early appearance of the virus in Italy.

### Materials and methods

Samples were tested for influenza [RT-PCR assay (A/H1N1, A/H3N2, B/Yam, B/Vic)] in accordance with the DRIVE study protocol. Subsequently, swabs underwent molecular testing for SARS-COV-2. [one-step real-time multiplex retro-transcription (RT) PCR].

**Funding:** The author(s) received no specific funding for this work.

**Competing interests:** The authors have declared that no competing interests exist.

## Results

In the 1683 samples collected, no evidence of SARS-CoV-2 was found. Moreover, 28.3% (477/1683) of swabs were positive for influenza viruses, the majority being type A (358 vs 119 type B). A/H3N2 was predominant among influenza A viruses (55%); among influenza B viruses, B/Victoria was prevalent. The highest influenza incidence rate was reported in patients aged 0–17 years (40.3%) followed by those aged 18–64 years (24.4%) and $\geq$65 years (14.8%).

## Conclusions

In Italy, some studies have shown the early circulation of SARS-CoV-2 in northern regions, those most severely affected during phase I of the pandemic. In central and southern regions, by contrast no early circulation of the virus was registered. These results are in line with ours. These findings highlight the need to continue to carry out retrospective studies, in order to understand the epidemiology of the novel coronavirus, to better identify the clinical characteristics of COVID-19 in comparison with other acute respiratory illnesses (ARI), and to evaluate the real burden of COVID-19 on the healthcare system.

## Introduction

On December 31, 2019, the Wuhan Municipal Health Commission in Wuhan City, Hubei province, China, reported a cluster of 27 pneumonia cases of unknown aetiology [1]. On January 9, 2020, the Chinese CDC stated that a novel coronavirus (later named SARS-CoV-2, the virus causing COVID-19) had been detected as the causative agent of 15 cases of pneumonia [2, 3]. On 11 March 2020, after assessing the levels of spread and severity of the SARS-CoV-2 infection, the World Health Organization (WHO) defined the COVID-19 outbreak as a pandemic [4].

The first European case was officially reported by France on January 24, 2020 [5]. One week later, in Italy, the first cases were described. These involved two Chinese tourists from Wuhan, who had landed in Milan and then fell ill in Rome on January 30, 2020. These patients were immediately put into isolation and are not believed to have infected anyone else [6]. The first autochthonous patient, a 38-year-old man, was diagnosed only one month later in Codogno (Lombardy), on February 21, 2020. It was believed to be the "patient zero", however when the virus was first introduced into Italy remains unclear. As, identifying the first introduction of the virus is of epidemiological interest in order to acquire new information on spread of SARS-CoV-2, many European countries have been trying to ascertain whether SARS-CoV-2 infections had occurred before the official first case reported by health authorities [7–13].

In this regard, we conducted a retrospective observational study to detect SARS-CoV-2 in oropharyngeal samples collected from hospitalized patients with Severe Acute Respiratory Infection (SARI) [14] aged $\geq$6 months in five hospitals in four Italian cities (Genoa, Rome, Bari, Siena) in the period 1$^{st}$ November 2019 – 29$^{th}$ February 2020. Our intention was to acquire new information on the real trend of the infection during phase I of the epidemic, and to determine the possible early appearance of the virus in Italy.

## Materials and methods

### Study population and period

During the 2019–2020 influenza season, oropharyngeal swabs were collected between 1$^{st}$ November 2019 and 29$^{th}$ February 2020 from hospitalized individuals with SARI aged $\geq$6 months enrolled in the European study DRIVE (Development of Robust and Innovative Vaccine Effectiveness) [15]. Data were collected through a network of hospitals (IT-BIVE-HOSP) composed of large academic tertiary hospitals with 400 to over 1,200 beds, located in:

- Liguria Region (North Italy)—San Martino Hospital is located in metropolitan area of Genoa, a city of 650,000 inhabitants. It is a tertiary teaching hospital with 1,200 beds and has more than 70 wards. The hospital is the acute care regional reference center for adults and accounts for 55% of all hospital admission in the metropolitan area.

- Tuscany Region (Central Italy)—Santa Maria alle Scotte Hospital is located in Siena. The hospital's catchment area is approximately of 120,000 inhabitants and has currently 700 beds.

- Lazio Region (Central Italy)–Two hospitals located in Rome (4,356,000 inhabitants) were involved in the network. Sant'Andrea Hospital is a university hospital for adults. It has 450 beds and provides 1,300,000 services per year (for both inpatients and outpatients). The Bambino Gesù Children's Hospital is the largest pediatric research hospital in Europe. It accounts 600 beds. The number of patients treated is very large with over 1,690,000 services every year for children and young people all over the world.

- Puglia Region (South Italy)- The Policlinico of Bari is a tertiary care referral hospital in the province of Bari (1,262,000 inhabitants) and one of the largest teaching hospitals in Southern Italy. The hospital has over 1,500 beds.

The study population's inclusion and exclusion criteria are described in the DRIVE protocol [15]. The demographic characteristics, chronic conditions, risk factors and influenza vaccination status of all patients were collected by means of a standardized questionnaire. Furthermore, clinical manifestations were also recorded by consulting medical records.

### SARI and COVID-19 definition

According to the European Center for Disease Control (ECDC) case definition, a case of SARI is defined as a hospitalized patient of any age with at least one respiratory sign or symptom (cough, sore throat, breathing difficulties) and at least one systemic sign or symptom (fever or low-grade fever, headache, myalgia, generalized malaise) or deterioration in general condition (asthenia, weight loss, anorexia or confusion and dizziness) [14].

A case of suspected COVID-19 is any person with at least one symptom such as cough, fever, shortness of breath, sudden onset of anosmia, ageusia or dysgeusia. Additional less specific symptoms may include headache, chills, muscle pain, fatigue, vomiting and/or diarrhoea [16].

### Molecular analysis for influenza detection

As the aim of the DRIVE study was to evaluate influenza vaccine effectiveness, all samples were tested for influenza viruses by means of the molecular method within 24 hours after collection. Total viral RNA was extracted from each respiratory swab and set up for PCR by means of the Nimbus IVD Seegene platform (STARMag 96x4 Viral DNA/RNA Universal Kit) using the Respiratory Panel 1-2-3 Assay kit (Seegene, Korea), according to the manufacturer's

instructions. The material extracted was tested to identify influenza (A/H1N1, A/H3N2, B/Yam, B/Vic) by means of a one-step real-time multiplex retro-transcription (RT) PCR assay on a Biorad CFX96™ thermal cycler. Three positive controls (one for each respiratory panel) and one internal control for viruses (common to all respiratory panels) were used for the analysis (included in the Seegene kit). Samples showing a cycle threshold (Ct) value <40 were considered positive [17]. All sample aliquots were stored at -20˚C.

## Molecular analysis for SARS-CoV-2 detection

Subsequently, swabs underwent molecular testing for SARS-COV-2. Total RNA was re-extracted from each respiratory swab and set up for PCR by means of the Nimbus IVD Seegene platform (STARMag 96x4 Viral DNA/RNA 200C Kit) using the Allplex™ 2019-nCoV Assay kit (Seegene, Korea), according to the manufacturer's instructions.

To verify the integrity of the RNA of the virus, a pool of 100 samples that had proved positive for influenza viruses were re-tested. The results obtained demonstrated the absence of RNA degradation in the samples. Moreover, further verification was carried out by inserting into the PCR analysis a human gene (human Rnase P) used to confirm the correctness of RNA extraction.

The material extracted was tested for the identification of SARS-COV-2 by means of a one-step real-time multiplex retro-transcription (RT) PCR assay on a Biorad CFX96™ thermal cycler, targeting the nucleoprotein region (N), RNA-dependent RNA-polymerase region (RdRp) and the envelope region (E). One positive control and one internal control were used for the analysis (included in the Seegene kit).

Samples showing a cycle threshold (Ct) value <40 were considered positive [18].

## Ethics statement

The study was performed in accordance with the World Medical Association's Declaration of Helsinki and the retrospective data were fully anonymized. The study protocol was approved by the Ethics Committee of the Liguria Region (Genoa, Italy) (n˚ 245/2019) as coordinator center and subsequently approved by all local the Ethics Committees. Informed written consent was obtained from each patient, as required by the DRIVE study protocol [15].

## Results

Overall, 1,683 hospitalized patients with SARI were enrolled at different times during the study period. Fig 1 shows the distribution of patients (= swabs) by week during the study period.

Data on demographic characteristics, chronic conditions, risk factors and influenza vaccination status were collected for every patient (Table 1). The patients' mean age was 38.2 years and about 35.7% (600/1,683) were ≥65 years old.

Clinical manifestations were also recorded for every patient (Table 2). The most common symptoms were fever (81.1%, 1,365/1683) and cough (60.1%, 1,012/1683) (Table 2); these are generic symptoms that could hypothetically be related to SARS-CoV-2 infection.

No evidence of SARS-CoV-2 was found in our retrospective study.

In accordance with the DRIVE protocol, we tested all swabs for influenza viruses. Overall, 28.3% (477/1,683) of our swabs were positive for influenza viruses: 358 (75%, 358/477) were type A and 119 (25%, 119/477) type B. The details of positive samples are shown in Fig 2. Most positive influenza cases were of subtype A/H3N2 and mainly affected subjects aged <18 years and the elderly. By contrast, subtype A/H1N1 was prevalent in adults (aged 18–64 years),

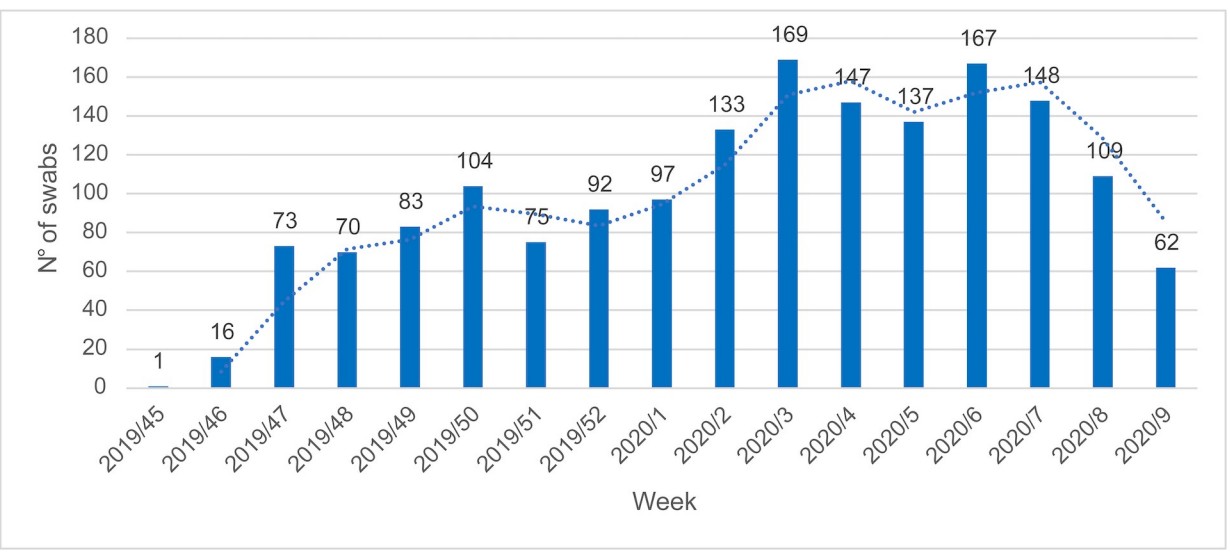

**Fig 1. Number of swabs distribution by week during the study period.**

followed by the elderly (Fig 2). 89.9% of influenza type B viruses were detected in subjects aged <18 years.

Subjects vaccinated against influenza in the 2019–2020 season were 18.9% (318/1,683). Specifically, 79.9% (254/318) were the elderly, 11.6% (37/318) were aged 18–64 years and 8.5% (27/318) were aged <18 years.

## Discussion

In Italy, the first official autochthonous case was diagnosed in Codogno (Lombardy), on February 21, 2020. This patient had been in contact with a colleague who had returned from a business trip to China. As the colleague tested negative for SARS-CoV-2, the first introduction of the virus into Italy remains unclear [10]. Over the following days, other cases were reported from several different areas of the country, with Northern Italy being most severely affected at the beginning of the COVID-19 pandemic [19, 20]. Identifying the first introduction of the

**Table 1. Patients' characteristics stratified by age.**

|  | <18 y | 18–64 y | ≥65 y |
|---|---|---|---|
| Total | 780 (100%) | 303 (100%) | 600 (100%) |
| Sex = male | 422 (54.1%) | 175 (57.8%) | 331 (55.2%) |
| **Any chronic condition*** | | | |
| No (0) | 666 (85.4%) | 174 (57.4%) | 95 (15.8%) |
| Yes (≥1) | 114 (14.6%) | 129 (42.6%) | 505 (84.2%) |
| **Influenza vaccination status (2019–2020 season)** | | | |
| No | 751 (96.3%) | 265 (87.5%) | 345 (57.5%) |
| Yes | 27 (3.5%) | 37 (12.2%) | 254 (42.3%) |
| N/A** | 2 (0.2%) | 1 (0.3%) | 1 (0.2%) |

* Chronic respiratory diseases, Heart or cardiovascular disease, Diabetes, Renal disease, Anemia, Cancer, Chronic liver disease, Dementia, History of stroke, Obesity, Autoimmune disease, Rheumatological diseases.

** Data not available.

**Table 2. Patients' clinical manifestations.**

| Symptoms | |
|---|---|
| Fever | 1,365 (81.1%) |
| Malaise | 798 (47.4%) |
| Headache | 196 (11.6%) |
| Myalgia | 271 (16.1%) |
| Cough | 1,012 (60.1%) |
| Sore throat | 731 (43.4%) |
| Short breath | 590 (35.1%) |

virus is of epidemiological importance in order to track and map the spread of SARS-CoV-2 in a country. For this reason, we retrospectively analyzed samples collected from hospitalized patients with SARI, as it has been demonstrated that patients with COVID-19 are more likely to be admitted to hospital; therefore, these patients were best suited to the aim of the study [21]. Moreover, the definition of COVID-19 overlaps with that of SARI, confirming that the clinical picture is insufficient in order to diagnose SARS-CoV-2 infection [21].

We found no evidence of SARS-CoV-2. Our results are in line with some Italian and European data. Capalbo et al. [13] evaluated the prevalence of SARS-CoV-2 infection among SARI patients in a hospital in Central Italy from November 1, 2019 to March 1, 2020. Like us, they confirmed that SARS-CoV-2 was not circulating at the time of their study and that the COVID-19 pandemic did not start before its official onset in Italy. Moreover, in a study conducted in Parma in the winter season 2020, Calderaro et al. did not find positive SARS-CoV-2

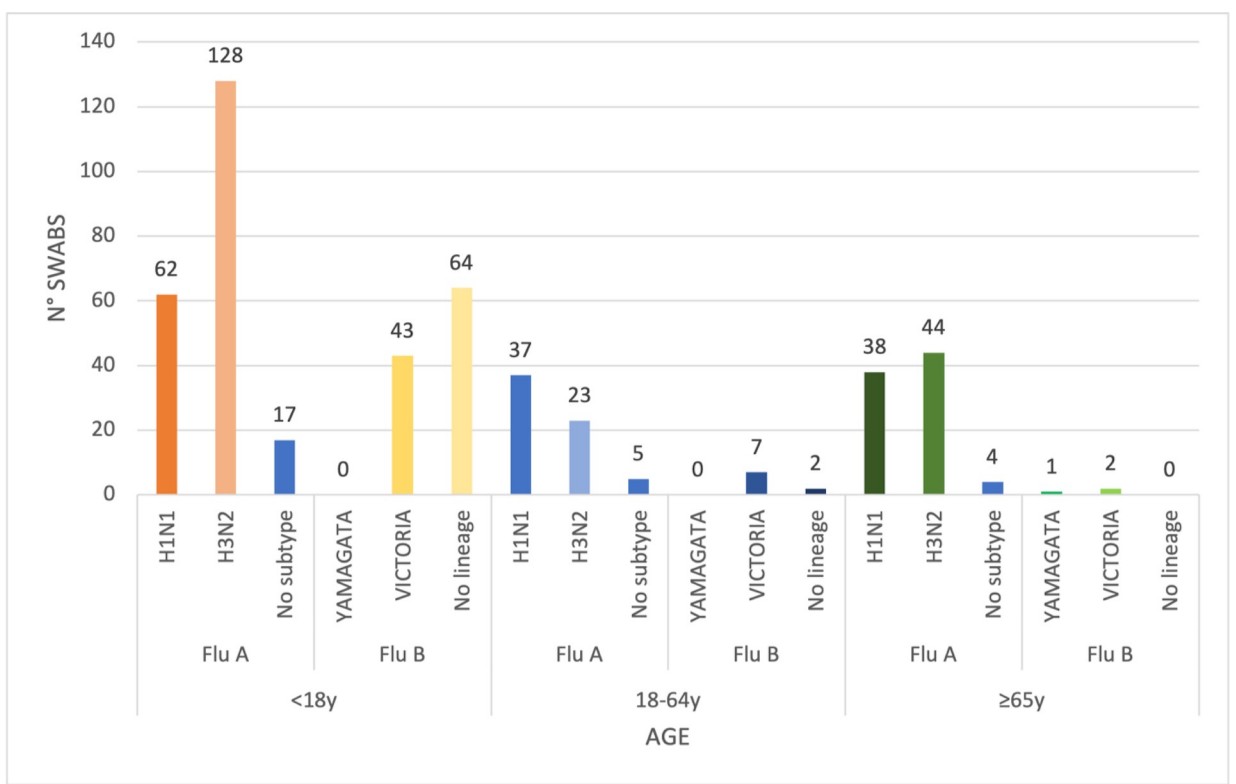

**Fig 2. Positive sample distribution by influenza type/subtype or lineage and age.**

cases in hospitalized children and adult patients. These authors reported that SARS-CoV-2 was absent in the study area until February 26, 2020 [22]. By contrast, other Italian studies revealed the presence of the virus before to the first "official" case. Some retrospective studies conducted on samples of a different sort (serological and environmental samples) and from patients with different health conditions reported different results from ours [7–10]. In Northern Italy, environmental waste-water monitoring detected positive samples as early as December 2019 [9, 10]. In Milan, Amendola et al. detected the RNA of the virus in early December 2019 in a swab sample from a child with suspected measles [8]. Finally, Apolone et al. found seroprevalence evidence of SARS-CoV-2 in asymptomatic patients. These were lifelong smokers who were screened for the early detection of lung cancer (high-risk group) from September to October 2019 [7]. It should be noted that the published Italian studies reporting early SARS-CoV-2 circulation were conducted in geographic areas that were different from ours, and which were severely affected during the initial phase I of the pandemic. This fact could explain the differences from our study.

In Europe, different results have been reported in different geographic area. For example, Tomb et al. detected no SARS-CoV-2 in Scotland prior to March 2020 [12], whereas in France, the results of the retrospective analysis conducted by Deslandes et al. on nasopharyngeal swabs collected from hospitalized patients suggested that the epidemic had probably started there in early December 2019 [11]. Notably, the first European case was officially reported by France on January 24, 2020 [5].

What is common to all these studies is the observation that the COVID-19 pandemic impacted influenza circulation from week 13 of 2020, when countries implemented strict lockdowns and issued hygiene recommendations [20, 23]. In line with the European trend, Italy's 2019–2020 influenza season had a shorter overall duration than previous seasons. Our study confirmed this trend; indeed, after the first week of March 2020, samples positive for influenza decreased drastically in all age-groups, and other respiratory pathogens were also rarely found. In addition, Calderaro et al. [22] pointed out that, from March 2020 onwards, SARS-CoV-2 became the main circulating respiratory pathogen, underlining the strong epidemic power of this coronavirus. These authors also reported that SARS-CoV-2 was found in mixed infections in only three cases [22].

Like all the literature studies considered, ours has some limitations. We used oropharyngeal swabs, while it has been shown that nasopharyngeal swabs are the most suitable for the molecular detection of SARS-CoV-2, as the quantity of virus is greater in the nose [24]. Moreover, although the RT-PCR assay is the gold standard for SARS-CoV-2 diagnosis, factors such as the sampling modality and the timing of sampling in relation to symptom onset might have modified the presence of viral RNA in the samples and reduced the sensitivity of the test. Finally, we stocked the swabs at -20˚C and later processed them for SARS-CoV-2 molecular detection. Although we followed all the protocols in order to minimize the possible degradation of the genetic material of any pathogens present in the samples during the phases of freezing and storage, it is known that the defrosting step can affect the results of the extraction and real-time steps. The tests carried out in order to detect possible RNA degradation confirmed the correctness of the procedures for the conservation of the samples and demonstrated that no degradation of the genetic material had taken place. Indeed, the results obtained from the second analysis of a pool of samples positive for influenza confirmed the results of the first analysis.

## Conclusions

Our results and the literature data show that it is very difficult to establish the exact time and place of the initial SARS-CoV-2 outbreak in Italy and Europe, highlighting the need to

continue to carry out retrospective studies in order to understand the epidemiology of the novel coronavirus, to better identify the clinical characteristics of COVID-19 in comparison with other acute respiratory illnesses (ARI), and to evaluate the real burden of COVID-19 on the healthcare system.

In sum, it is crucial to strengthen routine monitoring (both epidemiological and laboratory) of the causative agents of SARI, in order to support preventive strategies for all respiratory pathogens and promote integrated strategies for influenza and COVID-19 vaccination.

## Supporting information

**S1 File.**
(XLSX)

## Acknowledgments

All authors have read and agree to the published version of the manuscript. Conceptualization: G.I., D.P., A.O., C.N., I.M., C.R., M.C; data collection: B.M.P., P.L.L., E.B., D.L., E.P., M.L.C.D. A., M.R., C.R.; laboratory analyses: B.M.P., B.B., P.C., I.M., E.B., F.C., D.L., C.C., A.O.M, G.L.; interpretation of the data: P.L.L., A.O., D.P., G.B.O., M.C.; software, S.M.; drafting of the paper: D.P., B.M.P., A.O.; critical revision of manuscript: C.R., M.C., I.M., C.N., G.I., supervision, G.I.

## Author Contributions

**Conceptualization:** Donatella Panatto, Andrea Orsi, Christian Napoli, Ilaria Manini, Caterina Rizzo, Maria Chironna, Giancarlo Icardi.

**Data curation:** Donatella Panatto, Andrea Orsi, Beatrice Marina Pennati, Piero Luigi Lai, Bianca Bruzzone, Patrizia Caligiuri, Christian Napoli, Enrico Bertamino, Giovanni Battista Orsi, Daniela Loconsole, Francesca Centrone, Elisabetta Pandolfi, Marta Luisa Ciofi Degli Atti, Carlo Concato, Caterina Rizzo, Maria Chironna.

**Formal analysis:** Piero Luigi Lai, Stefano Mosca, Bianca Bruzzone, Patrizia Caligiuri, Christian Napoli, Giovanni Battista Orsi, Daniela Loconsole, Elisabetta Pandolfi, Marta Luisa Ciofi Degli Atti.

**Methodology:** Beatrice Marina Pennati, Bianca Bruzzone, Patrizia Caligiuri, Enrico Bertamino, Ilaria Manini, Daniela Loconsole, Francesca Centrone, Carlo Concato, Giulia Linardos, Andrea Onetti Muda, Massimiliano Raponi, Livia Piccioni.

**Software:** Piero Luigi Lai, Stefano Mosca.

**Supervision:** Giancarlo Icardi.

**Writing – original draft:** Donatella Panatto, Andrea Orsi, Beatrice Marina Pennati.

**Writing – review & editing:** Christian Napoli, Giovanni Battista Orsi, Ilaria Manini, Caterina Rizzo, Maria Chironna, Giancarlo Icardi.

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
