## [Decision Letter · Decision Letter 0]

24 May 2021

PONE-D-21-10636

No evidence of SARS-CoV-2 in hospitalized patients with severe acute respiratory
syndrome in five Italian hospitals from 1st November 2019 to 29th February 2020

PLOS ONE

Dear Dr. Panatto,

Thank you for submitting your manuscript to PLOS ONE. After careful consideration, we
feel that it has merit but does not fully meet PLOS ONE’s publication criteria as it
currently stands. Therefore, we invite you to submit a revised version of the
manuscript that addresses the points raised during the review process.

Please submit your revised manuscript by Jun 27 2021 11:59PM. If you will need more
time than this to complete your revisions, please reply to this message or contact
the journal office at plosone@plos.org. When
you're ready to submit your revision, log on to https://www.editorialmanager.com/pone/ and select the 'Submissions
Needing Revision' folder to locate your manuscript file.

Please refer to the detailed comments from Reviewer 1 in the attachment for your
revision.

If you would like to make changes to your financial disclosure, please include your
updated statement in your cover letter. Guidelines for resubmitting your figure
files are available below the reviewer comments at the end of this letter.

We look forward to receiving your revised manuscript.

Kind regards,

Wenbin Tan

Academic Editor

PLOS ONE

Journal Requirements:

2)  Thank you for including your ethics statement: "The study protocol was approved
by the Ethics Committee of the Liguria Region (Genoa, Italy) (n° 245/2019)."

a) Please provide additional details regarding participant consent. In the ethics
statement in the Methods and online submission information, please ensure that you
have specified (1) whether consent was informed and (2) what type you obtained (for
instance, written or verbal, and if verbal, how it was documented and witnessed). If
your study included minors, state whether you obtained consent from parents or
guardians. If the need for consent was waived by the ethics committee, please
include this information.

3) In the Methods section and the online ethics statement. Please provide some
clarification whether the current ethics statement provided is for the DRIVE study
or the current study reported.

4)  Please include in your Methods section (or in Supplementary Information files)
the participating hospitals/institutions.

Reviewers' comments:

Reviewer #1: All the sections of the manuscript need to be rewritten or to be
rephrased with more focus on the aim of the work and the value of the results. the
sections need to be better connected to show the importance of the findings. There
are some drawbacks in the sampling and storing. These drawbacks should be discussed
to clearly show their effects on the results and their reliability.

Reviewer #2: Please address the following points:

1. Were the tests for all the viruses done simultaneously? The authors have to rule
out the possibility that RNA in the samples was not degraded to yield a positive
test for SARS-CoV-2.

2. Are the authors sure that the RT-PCR tests were real time?

3. Which of the four cities are located in Northern Italy (the worst affected
region)?

4. ECDC? European Center for Disease Control?

5. Table 1: What does N/A mean? Explain.

6. Write figure legends or at least their titles.

---

## [Author Response · Author response to Decision Letter 0]

2 Sep 2021

Journal requirements

1) The manuscript meets PLOS ONE’s style requirements.

2) The sentence “The study protocol was approved by the Ethics Committee of the
Liguria Region (Genoa, Italy) (n◦ 245/2019) as coordinator center and subsequently
approved by all local the Ethics Committees” has been added.

a) In the manuscript, the paragraph “Ethics Statement”, containing the EC approval
code and the detail on informed consent, has been added.

3) The sentence “Informed written consent was obtained from each patient, as required
by the DRIVE study protocol [15]” has been added.

4) In the Methods section, the description of the hospitals has been added.

Reviewers’ comments

Reviewer #1: 

All the sections of the manuscript need to be rewritten or to be rephrased with more
focus on the aim of the work and the value of the results. the sections need to be
better connected to show the importance of the findings. There are some drawbacks in
the sampling and storing. These drawbacks should be discussed to clearly show their
effects on the results and their reliability.

Done

As required by reviewer 1, all sections have been rewritten in order to better focus
on the aims and results of the study.

All revisions are well visible in the revised manuscript with track change.

Reviewer #2: 

1) Were the tests for all the viruses done simultaneously? The authors have to rule
out the possibility that RNA in the samples was not degraded to yield a positive
test for SARS-CoV-2.

The text has been modified in order to better explain that the molecular analyses for
influenza and SARS-CoV-2 were not performed simultaneously. The molecular test for
influenza was performed before the test for SARS-CoV-2. The study biases have been
better explained.

2) Are the authors sure that the RT-PCR tests were real time?

Yes, all molecular tests were performed by means of the real-time method.

3) Which of the four cities are located in Northern Italy (the worst affected
region)?

Genoa is the only city located in Northern Italy

4) ECDC? European Center for Disease Control?

Yes

5) Table 1: What does N/A mean? Explain.

The notation N/A (not available) has been explained

6) Write figure legends or at least their titles.

Done

References have been updated and renumbered.

Figure 2 has been improved.

---

## [Decision Letter · Decision Letter 1]

23 Sep 2021

PONE-D-21-10636R1No evidence of SARS-CoV-2
in hospitalized patients with severe acute respiratory syndrome in five Italian
hospitals from 1st November 2019 to 29th February
2020PLOS ONE

Dear Dr. Panatto,

Thank you for submitting your manuscript to PLOS ONE. After careful consideration, we
feel that it has merit but does not fully meet PLOS ONE’s publication criteria as it
currently stands. Therefore, we invite you to submit a revised version of the
manuscript that addresses the points raised during the review process.

Please submit your revised manuscript by Nov 07 2021 11:59PM. If you will need more
time than this to complete your revisions, please reply to this message or contact
the journal office at plosone@plos.org. When
you're ready to submit your revision, log on to https://www.editorialmanager.com/pone/ and select the 'Submissions
Needing Revision' folder to locate your manuscript file.

Please include the following items when submitting your revised
manuscript:A rebuttal letter that responds to each point raised by the academic
editor and reviewer(s). You should upload this letter as a separate file
labeled 'Response to Reviewers'.A marked-up copy of your manuscript that highlights changes made to the
original version. You should upload this as a separate file labeled
'Revised Manuscript with Track Changes'.An unmarked version of your revised paper without tracked changes. You
should upload this as a separate file labeled 'Manuscript'.

If you would like to make changes to your financial disclosure, please include your
updated statement in your cover letter. Guidelines for resubmitting your figure
files are available below the reviewer comments at the end of this letter.

We look forward to receiving your revised manuscript.

Kind regards,

Wenbin Tan

Academic Editor

PLOS ONE

Journal Requirements:

Review Comments to the Author

Reviewer #1: All the comments have been addressed. I recommend the authors to rewrite
the conclusion section

Reviewer #2: The authors have indicated that the detection for SARS-CoV2 was done on
archived samples that were previously tested for Influenza viruses. They found that
all the samples were found negative for the coronavirus. The onus is on the
researchers to show that the RNA in the samples was not degraded. They should have
used some positive controls for these samples. For example test again for the
already detected Influenza virus, for example. Furthermore, they should indicate
whether they re-extracted RNA from the samples or used already extracted RNA from
these samples. These are extremely important points. For this reason, major revision
is indicated.

---

## [Author Response · Author response to Decision Letter 1]

9 Nov 2021

Genoa, 4 November 2021

Dear Academic Editor PLOS ONE,

Thank you for giving us the opportunity to resubmit our manuscript “No evidence of
SARS-CoV-2 in hospitalized patients with severe acute respiratory syndrome in five
Italian hospitals from 1st November 2019 to 29th February 2020” (PONE-D-21-10636)
for publication in your journal. The reviewers' comments were insightful and
pertinent.

Below, you will find the reviewers' reports with point-by-point replies and
explanations of the modifications made. The reviewers’ comments are in bold, while
our replies are in italics.

The revised manuscript has been uploaded to the system in two versions: one with
“track changes” and the other without “track changes”.

Sincerely,

Prof. Donatella Panatto

Reviewers’ comments

Reviewer #1: 

All the comments have been addressed. I recommend the authors to rewrite the
conclusion section

Done

Reviewer #2: 

The authors have indicated that the detection for SARS-CoV2 was done on archived
samples that were previously tested for Influenza viruses. They found that all the
samples were found negative for the coronavirus. The onus is on the researchers to
show that the RNA in the samples was not degraded. They should have used some
positive controls for these samples. For example test again for the already detected
Influenza virus. Furthermore, they should indicate whether they re-extracted RNA
from the samples or used already extracted RNA from these samples. These are
extremely important points. For this reason, major revision is indicated.

As required by reviewer 2, we have better specified, in the Materials and Methods
section and Discussion section, the method of verifying the correct re-extraction of
the genetic material from the sample and the integrity of the viral RNA.

---

## [Editor Report · Decision Letter 2]

22 Nov 2021

No evidence of SARS-CoV-2 in hospitalized patients with severe acute respiratory
syndrome in five Italian hospitals from 1st November 2019 to 29th February 2020

PONE-D-21-10636R2

Dear Dr. Panatto,

We’re pleased to inform you that your manuscript has been judged scientifically
suitable for publication and will be formally accepted for publication once it meets
all outstanding technical requirements.

Kind regards,

Wenbin Tan

Academic Editor

PLOS ONE

Additional Editor Comments:

The 2nd reviewer's comment was well addressed.

---

## [Editor Report · Acceptance letter]

25 Nov 2021

PONE-D-21-10636R2 

No evidence of SARS-CoV-2 in hospitalized patients with severe acute respiratory
syndrome in five Italian hospitals from 1^st^ November 2019 to
29^th^ February 2020 

Dear Dr. Panatto:

I'm pleased to inform you that your manuscript has been deemed suitable for
publication in PLOS ONE. Congratulations! Your manuscript is now with our production
department. 

Kind regards, 

on behalf of

Dr. Wenbin Tan 

Academic Editor

PLOS ONE